# Bidirectional Modulation of the Voltage-Gated Sodium (Nav1.6) Channel by Rationally Designed Peptidomimetics

**DOI:** 10.3390/molecules25153365

**Published:** 2020-07-24

**Authors:** Nolan M. Dvorak, Paul A. Wadsworth, Pingyuan Wang, Haiying Chen, Jia Zhou, Fernanda Laezza

**Affiliations:** 1Department of Pharmacology and Toxicology, University of Texas Medical Branch, Galveston, TX 75901, USA; nmdvorak@utmb.edu (N.M.D.); pawadswo@utmb.edu (P.A.W.); piwang@utmb.edu (P.W.); haichen@utmb.edu (H.C.); 2Center for Addiction Research, University of Texas Medical Branch, Galveston, TX 75901, USA

**Keywords:** protein:protein interactions (PPIs), voltage-gated Na^+^ (Nav) channels, fibroblast growth factor 14 (FGF14), peptidomimetics, molecular docking, neurotherapeutics

## Abstract

Disruption of protein:protein interactions (PPIs) that regulate the function of voltage-gated Na^+^ (Nav) channels leads to neural circuitry aberrations that have been implicated in numerous channelopathies. One example of this pathophysiology is mediated by dysfunction of the PPI between Nav1.6 and its regulatory protein fibroblast growth factor 14 (FGF14). Thus, peptides derived from FGF14 might exert modulatory actions on the FGF14:Nav1.6 complex that are functionally relevant. The tetrapeptide Glu-Tyr-Tyr-Val (EYYV) mimics surface residues of FGF14 at the β8–β9 loop, a structural region previously implicated in its binding to Nav1.6. Here, peptidomimetics derived from EYYV (**6**) were designed, synthesized, and pharmacologically evaluated to develop probes with improved potency. Addition of hydrophobic protective groups to **6** and truncation to a tripeptide (**12**) produced a potent inhibitor of FGF14:Nav1.6 complex assembly. Conversely, addition of hydrophobic protective groups to **6** followed by addition of an *N*-terminal benzoyl substituent (**19**) produced a potentiator of FGF14:Nav1.6 complex assembly. Subsequent functional evaluation using whole-cell patch-clamp electrophysiology confirmed their inverse activities, with **12** and **19** reducing and increasing Nav1.6-mediated transient current densities, respectively. Overall, we have identified a negative and positive allosteric modulator of Nav1.6, both of which could serve as scaffolds for the development of target-selective neurotherapeutics.

## 1. Introduction

In excitable cells, voltage-gated sodium (Nav) channels are the primary molecular determinants of the generation and conduction of action potentials [1]. Of the nine different pore-forming α-subunits that have been described (Nav1.1–1.9), the Nav1.1–1.3 and 1.6 isoforms are the primary Nav channels expressed in the central nervous system (CNS) [2]. Given their primacy in modulating neuronal excitability, it is unsurprising that aberrant activity of these isoforms has been implicated in the etiologies of numerous neurologic and neurodevelopmental disorders, including epilepsy [3,4,5,6], migraines [7,8,9], and autism [10,11,12]. Lending credence to the involvement of Nav channel aberrations in neuropsychiatric disorders, Nav channel blockers are also commonly used as adjunct therapies for the treatment of bipolar disorder [13,14], anxiety [15], and schizophrenia [16,17,18]. Given this primacy of Nav channel dysfunction in the etiology of virtually all CNS disorders, they have been the target of many drug discovery campaigns. Such efforts, however, have been of limited success, as developed lead compounds often fail to demonstrate appreciable isoform selectivity and resultantly confer undesirable off-target side effects [19].

Nav1.6 channels are abundantly expressed in medium spiny neurons (MSNs) of the nucleus accumbens (NAc), a neuroanatomical region that governs mesocorticolimbic circuitry and is involved in motivation [20], reward processing [21], learning [22], and locomotion [23,24,25]. With translational studies increasingly illuminating linkages between dysfunction of MSNs in the NAc and neuropsychiatric and neurological symptomologies [26,27,28,29], the development of chemical probes to modulate these neurons would enable further mechanistic elucidation of these neuropathophysiological processes. Given the abundant expression of Nav1.6 and its regulatory protein fibroblast growth factor 14 (FGF14) in these neurons, a feasible approach for developing such probes is through the rational design of small molecules targeting the FGF14:Nav1.6 protein:protein interaction (PPI) interface [30,31].

To that end, we previously identified two peptides capable of modulating FGF14:Nav1.6 complex assembly [32]. One of these peptides, Phe-Leu-Pro-Lys (FLPK; **1**, Figure 1A), was derived from a four amino acid sequence located on the β12 sheet of FGF14 at the FGF14:Nav1.6 PPI interface, whereas the other, Glu-Tyr-Tyr-Val (EYYV; **5**, Figure 1B), was derived from an amino acid sequence located on the exposed β8–β9 loop of FGF14 [32]. To improve the potency and drug-like properties of **1**, we acetylated and aminated its *N*-terminus and *C*-terminus respectively (**2**, Figure 1A), and subsequently designed, synthesized, and pharmacologically evaluated peptidomimetics derived from the protected FLPK scaffold [30,31]. Briefly, altering the *N*-terminal substituent to Cbz and adding a Boc protective group to the constituent lysine residue of the protected FLPK scaffold produced ZL181 (**4**, Figure 1A), which inhibited FGF14:Nav1.6 complex assembly (half maximal inhibitory concentration (IC_50_) = 63 µM) and decreased maximal and instantaneous firing frequencies in MSNs of the NAc [31]. Subsequent efforts to optimize **2** included altering the *C*-terminal substituent to methoxyl (OMe) and the addition of a Fmoc protective group to the constituent lysine residue of the protected FLPK scaffold, which produced ZL0177 (**3**, Figure 1A). Crucially, this analog displayed markedly improved potency, inhibiting FGF14:Nav1.6 complex assembly with an IC_50_ value of 11 µM. Additionally, mechanism of action (MOA) studies using whole-cell patch-clamp electrophysiology revealed that ZL0177 significantly reduced Nav1.6-mediated peak current density and caused a depolarizing shift in voltage-dependence of Nav1.6 channel activation, suggesting that the peptidomimetic’s activity is conferred by functioning as a partial FGF14 mimic [30]. 

To develop additional probes to further elucidate the intricate biophysical changes Nav1.6 channels undergo on account of their PPI with FGF14, we herein report our derivation and pharmacological evaluation of EYYV analogs (**6**, Figure 1B). To do so, all newly synthesized peptidomimetics were screened using an in-cell assay, which revealed that addition of hydrophobic protective groups to **6** followed by truncation to a tripeptide produced a potent inhibitor of FGF14:Nav1.6 complex assembly. Conversely, addition of hydrophobic protective groups to **6** followed by addition of an *N*-terminal benzoyl substituent produced a potentiator of the complex’s assembly. After surface plasmon resonance (SPR) studies revealed that both peptidomimetics exhibited great affinity for Nav1.6, subsequent functional evaluation of the two analogs using whole-cell patch-clamp electrophysiology confirmed their inverse activities, with the inhibitor and potentiator of FGF14:Nav1.6 complex assembly reducing and increasing Nav1.6-mediated transient current densities, respectively. Overall, by employing this novel approach that combines in-cell screening with orthogonal validation measures, including SPR and whole-cell patch-clamp electrophysiology, we were able to identify both a negative and positive allosteric modulator of Nav1.6, both of which could serve as ideal scaffolds for the development of targeted pharmacological probes and, with further chemical optimization, potential neurotherapeutic drug leads.

## 2. Results

### 2.1. Chemistry

As outlined in Scheme 1, the diverse EYYV analogs were synthesized with modifications to both their *C*- and *N*-termini (Figure 1B). The dipeptide analogs (**10** and **11**) were generated by condensation of commercially available compound **7** with compound **8** and **9**, respectively. Deprotecting the Fmoc group of compounds **8** and **9**, followed by coupling with compound **7**, led to tripeptide compounds **12** and **13**, respectively. Following a similar synthetic procedure to that of the preparation of compound **12**, compounds **12** and **13** were converted into tetrapeptide compounds **15** and **16** in excellent yields. Compounds **17**–**24** were prepared by deprotection of compounds **15** and **16** and the subsequent introduction of the corresponding substituents, including acetyl, benzoyl, adamantanyl carbamic, adamantane-1-carbonyl, decanoyl, cyclohexanecarbonyl, and Cbz. Copies of ^1^H and ^13^C NMR spectra of all newly synthesized analogs are included in the Appendix A.

### 2.2. Biology

#### 2.2.1. In-Cell Testing of Analogs Using the Split Luciferase Complementation Assay (LCA)

We previously developed, optimized, and reported an in-cell LCA that allows for the reproducible identification of modulators of FGF14:Nav1.6 complex assembly [33]. Briefly, a HEK293 cell line stably expressing both CLuc-FGF14 and CD4-Nav1.6-NLuc recombinant proteins (hereafter referred to as Clone V cells) was developed. In this system, binding of FGF14 to the *C*-terminal tail of Nav1.6 facilitates the reconstitution of the luciferase enzyme, which in the presence of the substrate luciferin, produces luminescence. This assay was employed in the present investigation by reconstituting all the newly synthesized analogs derived from **6** in dimethyl sulfoxide (DMSO), bringing them to a preliminary screening concentration of 25 µM, administering them to Clone V cells seeded into 96-well plates, and subsequently normalizing the luminescence values observed in treatment wells to per plate controls (0.5% DMSO alone), the results of which are summarized in Figure 2A and Table 1. Before proceeding with further analyses, however, we first ensured that the observed modulatory effects of these peptidomimetics were not merely artifacts stemming from cytotoxicity. To do so, all EYYV analogs were tested using the CellTiter-Blue^®^ Cell Viability (CTB) assay (Figure 2B), which revealed that none of the screened analogs demonstrated cytotoxicity (Figure 2B).

As shown in Figure 2A and Table 1, we first investigated how the addition of hydrophobic protective groups to, and truncation of, **6** to a dipeptide (**10**) and tripeptide (**12**) modulated in-cell pharmacological activity. Whereas truncation to a dipeptide resulted in a loss of activity, the tripeptide analog (**12**) markedly inhibited FGF14:Nav1.6 complex assembly (Figure 2A and Table 1) and displayed low micromolar potency (IC_50_ = 23.7 µM; Figure 3 and Table 2).

After characterizing the in-cell activity of the dipeptide and tripeptide analog, we subsequently sought to design, synthesize, and pharmacologically evaluate tetrapeptide analogs with improved cLogP values and stability compared to the parental EYYV peptide. To do so, hydrophobic protective groups were added to the protected EYYV scaffold, which produced **17**. In-cell screening of **17** revealed that it markedly potentiated assembly of the FGF14:Nav1.6 complex (Figure 2A and Table 1) and displayed low micromolar potency (EC_50_ = 12.5 µM; Figure 3 and Table 2). Having identified a potentiator of FGF14:Nav1.6 complex assembly, further chemical interventions were employed to develop a more efficacious enhancer. To do so, we first replaced the NH_2_ substituent at the *C*-terminus of compound **17** with a methoxyl (OMe) group, which produced **18**. This R^1^ modification to OMe resulted in a complete loss of in-cell activity, a recurring finding that was observed in the only other analog with a *C*-terminal OMe (**16**), suggesting that a *C*-terminal NH_2_ substituent played a crucial role in conferring analogs with in-cell activity. To investigate how altering the hydrophobicity of peptidomimetics modulated their pharmacological activity, the acetyl group at the *N*-terminus of **17** was modified to various lipophilic groups, which produced **15**, **19**, **20**, and **21**. Substituting the Ac group with a Fmoc group (**15**) resulted in a significant loss of in-cell activity, a phenomenon that is likely attributable to the substituent conferring the analog with a highly steric structure and unfavorable cLogP value (cLogP = 7.45; Table 1). Further efforts to optimize **17** by introducing a *N*-terminal aryl group produced **19**. Despite this analog displaying lessened potency relative to **17** (half maximal effective concentration (EC_50_) = 24.5 ± 1.7 µM versus 12.5 ± 1.9 µM for **19** and **17**, respectively; *p* < 0.005), its efficacy was significantly improved (E_max_ = 192.4% ± 3.2% versus 153.4% ± 3.5% for **19** and **17**, respectively; *p* < 0.0005; Figure 3 and Table 2). Whereas replacing the Ac group of **17** with an aryl group improved efficacy, modification of the R^2^ locale to fused alkyl ring substituents (**20** and **21**) abrogated in-cell activity. Additional attempts to optimize **17** entailed altering its *N*-terminal substituent to a long-chain alkyl, cycloalkyl, and Cbz group, which produced **22**, **23**, and **24**, respectively. Although **22** and **24** retained activity, **22** displayed lessened potency (EC_50_ = 47.5 µM; Figure 3 and Table 2) relative to **17**, and **24′**s potentiation of FGF14:Nav1.6 complex assembly was comparably mild (Figure 2A and Table 1). Overall, these SAR studies using the LCA identified three potential positive allosteric modulators (PAMs; **17**, **19**, and **22**) and one negative allosteric modulator (NAM; **12**) of FGF14:Nav1.6 complex assembly, the four of which were selected for protein:ligand binding studies using SPR.

#### 2.2.2. Characterization of Peptidomimetic Interactions with FGF14 and Nav1.6

After SAR studies using the LCA-identified potential PAMs (**17**, **19**, and **22**) and a NAM (**12**), we next sought to examine their kinetic interactions with FGF14 and Nav1.6 *C*-terminal tail. SPR-based kinetic analysis of interactions between drug-like compounds and target proteins has become a key method for drug discovery [34,35]. The determined k_on_ and k_off_ rate constants provide important information about interaction mechanisms and compound properties, enabling investigation of SARs and the rational modification of compounds. Therefore, we used SPR to assess and compare the binding of peptidomimetics to FGF14 and Nav1.6 *C*-terminal tail protein. Proteins were purified as previously described [31,36], immobilized to CM5 sensor chips, and increasing concentrations of compounds (0.195–100 µM) were flown over the chips at 60 µL/min. The results are shown in Figure 4 and Table 3 and Table 4. With the exception of **22**, all compounds bound appreciably to FGF14 with affinities ranging from 2.9 to 14.3 μM (Table 3, left). While **12** and **19** bound strongly to Nav1.6, with affinities ranging from 2.3 to 9.7 μM, lower affinities were observed for **17** and **22** (Table 3, right). 

Interestingly, the tripeptide (**12**) demonstrated two-fold greater binding affinity toward FGF14 compared to **19** (K_D_ of 2.98 vs. 6.49 µM, respectively; *p* < 0.05). As similar dissociation kinetics were observed for the two compounds (k_off_ = 1.86 × 10^−2^ and 1.30 × 10^−2^ s^−1^ for **12** and **19**, respectively; *p* > 0.4), this difference was largely driven by the faster association rate of **12**, which can also be observed by the leftward shift of the FGF14 steady-state saturation curve (red) for **12** relative to **19** (Figure 4, right-top and lower middle). The affinities of the other two tetrapeptide analogs, **17** and **22**, toward FGF14, were reduced by an even greater extent, collectively indicating that the presence of the glutamic acid residue, regardless of the R^2^ substituent present, prevents high-affinity FGF14 binding. This may suggest that the binding region of FGF14 is topologically constrained such that it cannot readily accommodate the larger tetrapeptide analogs, or that the presence of the glutamic acid residue masks crucial interactions between the YYV motif and FGF14.

Whereas the presence of the glutamic acid residue reduced the affinity of analogs toward FGF14, its inclusion conferred tighter binding to Nav1.6. This finding was underscored by the tripeptide exhibiting the lowest detectable binding strength toward Nav1.6. This suggests that the region of Nav1.6 bound to by these analogs is less topologically constrained relative to FGF14. Additionally, compared to **12** and **17**, **19** exhibited about 4-fold greater binding affinity toward Nav1.6 (*p* < 0.005 for **12** vs. **19**; *p* < 0.05 for **12** vs. **19**), a difference largely driven by its comparatively slow dissociation rate (k_off_ = 1.5 × 10^−1^ s^−1^). This difference can similarly be observed via the leftward shift of the Nav1.6 steady-state saturation curve (Figure 4, right, blue curve).

Consistent with the SAR studies using the LCA, which revealed that modification of **17′**s R^2^ substituent from Ac to benzoyl (**19**) improved efficacy (Table 2), this modification also resulted in an approximately 2-fold increase in binding to FGF14 and a 4-fold increase in binding to Nav1.6. Kinetically, this improved binding affinity was mediated via the benzoyl substitution conferring slower association and dissociation rates toward both proteins (as depicted by greater curvature in the SPR sensorgrams), a change likely attributable to the substituent’s constitutive π bonds enabling additional interactions with both binding partners. Overall, given **19′**s heightened efficacy and binding affinities toward both FGF14 and Nav1.6 relative to the other potential PAMs (**17** and **22**), it, along with the potential NAM (**12**), were selected for functional evaluation.

While it is potentially somewhat surprising that **12**, **17**, and **19** bind to both FGF14 and the *C*-terminal tail of Nav1.6, this finding is consistent with a previous investigation that revealed both overlap and structural divergence between the FGF14:FGF14 homodimer and FGF14:Nav1.6 complex PPI interfaces [36]. That these peptidomimetics bind to both FGF14 and Nav1.6 is desirable in that it affords multiple mechanisms by which they could modulate the PPI between FGF14 and Nav1.6. In the case of the inhibitor (**12**), its disruption of the PPI between FGF14 and Nav1.6 could be conferred via direct binding to the FGF14 interaction site on the *C*-terminal tail of Nav1.6 or via binding to FGF14 and causing the protein to undergo a conformational change that makes it inaccessible to its native interaction site on the *C*-terminal tail of Nav1.6. Similarly, for the potentiator of FGF14:Nav1.6 complex assembly (**19**), its modulatory effects on the PPI could be conferred by both binding to the *C*-terminal tail of Nav1.6 that makes the FGF14 interaction site increasingly accessible to the regulatory protein, or, conversely, by binding to FGF14 and causing it to undergo a conformational change that affords it with increased accessibility to its interaction site on the *C*-terminal tail of Nav1.6. In the native system, it is expected that the protein:ligand interactions of **12** and **19** with FGF14 and the *C*-terminal tail of Nav1.6 will concomitantly occur and collectively enable functionally relevant modulation of FGF14:Nav1.6 complex assembly.

#### 2.2.3. Electrophysiological Evaluation of Compounds **12** and **19**

Heretofore, it had been shown that addition of a *N*-terminal benzoyl substituent (**19**) to tetrapeptide analogs produced an efficacious potentiator of FGF14:Nav1.6 complex assembly, whereas truncation to a tripeptide (**12**) yielded an inhibitor of the complex′s assembly. Additionally, both **12** and **19** demonstrated promising protein:ligand binding interactions. As such, in an effort to further investigate their seemingly inverse activities, **12** and **19** were selected for functional evaluation as modulators of Nav1.6. To do so, HEK293 cells stably expressing Nav1.6 (HEK-Nav1.6) were incubated for 30 min with 0.1% DMSO (control), 20 µM 12, or 20 µM 19. After incubation, the modulatory effects of these peptidomimetics on Nav1.6-mediated currents were assessed using whole-cell patch-clamp electrophysiology (Figure 5 and Table 4). 

The first kinetic property of Nav1.6 channels we investigated as potentially being modulated by **12** and **19** was peak current density. Investigation of how this property was modulated by these analogs revealed that treatment with **12** (–18.30 ± 2.02 picoampere per picofarad (pA/pF); *n* = 10) significantly reduced Nav1.6-mediated peak current density derived from transient Na+ current relative to treatment with 0.1% DMSO (–55.39 ± 2.16 pA/pF; *n* = 10), whereas treatment with **19** had the inverse effect and potentiated peak current density (–79.63 ± 3.55 pA/pF; *n* = 8) (Figure 5B,C). The observed phenotypes could be attributable to the compounds altering channel availability by favoring or inhibiting steady-state (closed) inactivation, single channel conductance, or the number of channels at the plasma membrane. Such alterations could be mediated by **12** and **19** binding to the channel and inducing conformational changes that favor the channel adopting non-conductive (closed or inactivated) and conductive (open) states, respectively.

Lending further credence to **19′**s potentiation of Nav1.6 channel activity, subsequent investigations revealed that treatment with **19** induced a –20.03 ± 0.85 mV (0.1% DMSO; *n* = 10) to –22.49 ± 0.72 mV hyperpolarizing shift (*n* = 8; *p* < 0.05) of V_1/2_ of activation (Figure 5E,F), which was not accompanied by a change in the V_1/2_ of steady-state inactivation (*n* = 8; *p* = 0.984) (Figure 5G,H). Conversely, treatment with **12** affected neither the V_1/2_ of activation (*n* = 10; *p* = 0.756) nor the V_1/2_ of steady-state inactivation (*n* = 6; *p* = 0.600) relative to treatment with 0.1% DMSO (Figure 5E–H). Interestingly, both **12** and **19** were shown to significantly increase the long-term inactivation of Nav1.6 channels relative to the control (Figure 5I,J). Overall, these findings suggest that both peptidomimetics have complex mechanisms of action mediated via modulation of multiple kinetic properties of Nav1.6 channels that collectively confer them with inverse activities.

#### 2.2.4. Molecular Docking Studies of Compounds **12** and **19**

With compounds **12** and **19** functionally validated as negative and positive allosteric modulators of Nav1.6 respectively, we next docked these compounds to the Nav1.6 *C*-terminal tail homology model [36] in silico to further elucidate their binding modes using Schrödinger Small-Molecule Drug Discovery Suite. Both compounds **12** and **19** docked well into the Nav1.6 *C*-terminal tail at a binding site identical to the EYYV motif (within the β8–β9 loop) of FGF14, enabling interactions with the same group of key residues, as shown in Figure 6. For compound **12** (as depicted in Figure 6A,B), the constituent oxygen atom of the backbone carboynl group between Y and V forms hydrogen bonds with Arg1891. The NH_2_ group at the *C*-terminus of compound **12** forms two hydrogen bonds with residues Met1832 and Asp1833, whereas the Fmoc protecting group at the *N*-terminus of compound **12** interacts with Arg1866 via a π-cation interaction. 

The predicted docking model of compound **19** with the Nav1.6 *C*-terminal tail (Figure 6C,D) is fundamentally different from that of compound **12**, which may help elucidate the opposing actions of the two compounds on Nav1.6 activity. Whereas the *C*-terminus of **12** interacted strongly with Asp1833 and Met1832 via hydrogen bonds, compound **19** was inversely oriented, with its *C*-terminus engaging with Ser1859 and Asp1863. NH groups between the *N*-terminal benzoyl substituent and the glutamic acid residue of the backbone and between the two tyrosine residues of the backbone form hydrogen bonds with Ala1831 and Asp1858, respectively. Meanwhile, the constituent oxygen atom of the ester bond at the sidechain of the glutamic acid residue of the tetrapeptide backbone interacts with Arg1891 through hydrogen bonding. Like compound **12**, the NH_2_ group at the *C*-terminus of compound **19** also forms two hydrogen bonds, thus providing a structural rationale for why replacing the *C*-terminal NH_2_ with OMe abrogated the in-cell activity of analogs. The *N*-terminal benzoyl substituent of compound **19** forms hydrophobic interactions with a small hydrophobic pocket in the Nav1.6 *C*-terminal tail that is surrounded by Ile1827, Glu1828, Ile1830, Ala1831, and Met1832, a result in agreement with our SAR studies demonstrating that introducing a benzene ring at the *N*-terminus is essential for heightened efficacy.

Additionally, an overlay of the FGF14:Nav1.6 complex with the highest scoring binding poses for compounds **12**, **17**, and **19** (Figure 6E) exemplifies the drastic binding differences that result from the chemical changes, with the greatest disparity observed between **12** and **19**. Importantly, the docking results for the tripeptide (**12**) point toward its binding interactions most resembling those of the native surface residues on the β8–β9 loop of FGF14 at its interface with Nav1.6 (shown as orange in Figure 6E). Theses similarities between the binding interactions of **12** and FGF14 with the Nav1.6 *C*-terminal tail suggest that **12′**s truncated size relative to tetrapeptide analogs confers it with heightened mimicry of FGF14, thereby giving rise to its marked suppression of Nav1.6-mediated peak current density. Based upon theses structural analyses, and consistent with current models of Nav channel function [37,38,39], we propose that **12** and **19** exert opposite effects on Nav1.6-mediated currents as a result of differential interactions with the EF hand-like (EFL) and IQ domains of the *C*-terminal tail of the channel at sites that have established roles in channel trafficking and inactivation.

## 3. Discussion

Although it has been reported that aberrant firing of striatal neurons is implicated in a multitude of neuropsychiatric and neurological symptomologies [26,27,28,29,40,41], elucidation of how perturbations in their constituent PPIs contribute to these neuropathophysiological processes remains hampered by a lack of target-selective chemical probes. To that end, we sought to identify small molecule modulators of the PPI between FGF14 and Nav1.6, two proteins that are abundantly expressed in MSNs of the NAc and whose PPI, when perturbed, leads to neural circuitry aberrations [42,43]. To do so, we designed peptidomimetics derived from the EYYV parental tetrapeptide by employing a number of chemical interventions, including: (A) truncation of the tetrapeptide to a tripetide (YYV) and dipeptide (YV), (B) protection of the *C*-terminus of **6**, and (C) modification of the *N*-terminus of **6** with various substituents. After these newly designed peptidomimetics were synthesized, their modulatory effects on FGF14:Nav1.6 complex assembly were assessed using the LCA. In-cell screening of these analogs revealed that a tripeptide (**12**) displayed low micromolar inhibitory activity against FGF14:Nav1.6 complex assembly (IC_50_ = 23.7 µM), whereas addition of hydrophobic protective groups to **6** followed by addition of a N-terminal acetyl (**17**) or benzoyl (**19**) substituent produced potentiators of the complex’s assembly. Subsequent dose response analyses using the LCA revealed that **19** was the most efficacious analog among the potentiators. Protein:ligand binding studies of the four most promising hits revealed that **12** showed strong binding affinities toward both FGF14 and Nav1.6 *C*-terminal tail proteins, with K_D_ values of 2.88 and 9.86 µM, respectively. Additionally, **19** displayed appreciable binding affinities toward both of these proteins (K_D_ = 6.3 and 2.25 µM for FGF14 and Nav1.6, respectively). Functional evaluation of these two analogs using whole-cell patch-clamp electrophysiology confirmed their inverse activities, with **12** and **19** reducing and increasing Nav1.6-mediated transient current densities, respectively. Lastly, molecular docking studies of compounds **12** and **19** with the Nav1.6 *C*-terminal tail homology model were performed, which indicated that these two compounds formed crucial hydrogen bonds and hydrophobic interactions with Nav1.6.

The findings of the present investigation are consistent with previous efforts to develop small molecule modulators of PPIs between Nav channels and their regulatory accessory proteins [30,31], although the results differ in subtle, but crucial ways. For example, we previously presented ZL0177 [30], a novel peptidomimetic derived from FLPK, a four amino acid sequence located on the β12 sheet of FGF14 at its interface with Nav1.6 [32]. Like the herein reported compound **12**, ZL0177 suppressed Na_v_1.6-mediated peak current density; however, it also accelerated the kinetics of current decay and induced a depolarizing shift in the voltage-dependence of Na_v_1.6 channel activation [30]. That these phenotypes are not also induced by compound **12,** despite **12** inducing an even stronger suppression of Na_v_1.6-mediated peak current density relative to ZL0177 (−18.30 ± 2.02 pA/pF vs. –26.65 ± 6.3 pA/pF for **12** and ZL0177, respectively) likely stems from **12** being derived from EYYV, which, unlike FLPK, is located on the exposed β8–β9 loop of FGF14 at its interface with the *C*-terminal tail of Nav1.6. As such, it is expected that **12** and ZL0177 will differentially bind to the FGF14 interaction site on the *C*-terminal tail of Na_v_1.6 and induce slightly divergent phenotypes.

Currently, these peptidomimetics are being employed as novel pharmacological probes to interrogate the biophysical properties of Nav channels. Given their high molecular weights, as well as their high clogP and tPSA values, chemical optimization of these peptidomimetics by replacing their natural amino acids with non-natural amino acids and substituting functional groups to confer improved water solubility is likely necessary to permit blood-brain permeability and will be the subject of future investigations [44]. By pursuing these chemical optimization strategies, we envision that the negative and positive allosteric modulator of Nav1.6 herein identified could be developed into CNS penetrant lead compounds with promising potential to be advanced into latter stages of preclinical testing as PPI-based neurotherapeutics.

## 4. Materials and Methods

### 4.1. Chemistry

#### 4.1.1. General

All commercially available starting materials and solvents were reagent grade and used without further purification. Reactions were performed under a nitrogen atmosphere in dry glassware with magnetic stirring. Preparative column chromatography was performed using silica gel 60, particle size 0.063–0.200 mm (70–230 mesh, flash). Analytical thin-layer chromatography (TLC) was carried out employing silica gel 60 F254 plates (Merck, Darmstadt, Germany). Visualization of the developed chromatograms was performed with detection by ultraviolet (UV) light (254 nm). NMR spectra were recorded on a Brucker-300 (^1^H, 300 MHz; ^13^C, 75 MHz; Brucker, Billerica, MA, USA) spectrometer. ^1^H and ^13^C NMR spectra were recorded with tetramethylsilane (TMS) as an internal reference. Chemical shifts were expressed in ppm, and *J* values were given in Hz. High-resolution mass spectra (HRMS) were obtained from a Thermo Fisher LTQ Orbitrap Elite mass spectrometer (Thermo Fischer Scientific, Grand Island, NY, USA) Parameters include the following: Nano electrospray ionization (ESI) spray voltage was 1.8 kV, capillary temperature was 275 °C, and the resolution was 60,000, ionization was achieved by positive mode. Purities of final compounds were established by analytical high-performance liquid chromatography (HPLC), which was carried out on a Shimadzu HPLC system (model: CBM-20A LC-20AD SPD-20A U*V/V*IS). HPLC analysis conditions: Waters μBondapak C18 (300 × 3.9 mm), flow rate 0.5 mL/min, UV detection at 270 and 254 nm, linear gradient from 10% acetonitrile in water to 100% acetonitrile in water. All biologically evaluated compounds are >95% pure.

#### 4.1.2. Synthesis of (9*H*-fluoren-9-yl)methyl ((*S*)-1-(((*S*)-1-amino-3-methyl-1-oxobutan-2-yl)amino)-3-(4-(tert-butoxy)phenyl)-1-oxopropan-2-yl)carbamate (**10**)

(*S*)-2-((((9*H*-fluoren-9-yl)methoxy)carbonyl)amino)-3-(4-(*tert*-butoxy)phenyl)propanoic acid (**7**) (1.4 g, 3 mmol) and (*S*)-2-amino-3-methylbutanamide hydrochloride (**8**) (459 mg, 3 mmol) were dissolved in 20 mL of CH_2_Cl_2_ and the mixture solution was cooled to 0 °C with an ice bath. 1-hydroxybenzotriazole (HOBt) (405 mg, 3 mmol), 2-(1*H*-Benzotriazole-1-yl)-1,1,3,3-tetramethyluronium hexafluorophosphate (HBTU) (2.3 mg, 6 mmol), and *N,N*-Diisopropylethylamine (DIPEA) (2 mL, 12 mmol) were added to the solution at 0 °C. Then, the ice bath was removed, and the mixture solution was stirred at room temperature overnight. After the reaction was completed (detected by TLC), the mixture was washed with 1 N NaHSO_4_, saturated in NaHCO_3_ and brine. After drying over anhydrous Na_2_SO_4_, the solution was concentrated and purified with silica gel column (CH_2_Cl_2_/MeOH = 50/1) to obtain **10** (1.8 g, 93%) as a white solid. ^1^H NMR (300 MHz, DMSO-*d*_6_) *δ* 7.87 (d, *J* = 7.5 Hz, 2H), 7.76 (d, *J* = 8.9 Hz, 1H), 7.68–7.63 (m, 2H), 7.41 (t, *J* = 7.4 Hz, 3H), 7.34-7.28 (m, 2H), 7.20 (d, *J* = 8.4 Hz, 2H), 7.06 (s, 1H), 6.80 (d, *J* = 8.3 Hz, 2H), 4.39–4.02 (m, 5H), 3.43–3.36 (m, 1H), 2.97 (dd, *J* = 13.8, 3.8 Hz, 1H), 2.75 (d, *J* = 12.3 Hz, 1H), 1.97 (h, *J* = 6.8 Hz, 1H), 1.18 (s, 9H), 0.85 (t, *J* = 6.4 Hz, 6H). ^13^C NMR (75 MHz, DMSO) *δ* 173.16, 171.76, 156.22, 153.81, 144.15, 141.10, 133.13, 130.16, 128.06, 127.50, 125.73, 123.69, 120.50, 77.99, 66.16, 57.74, 56.71, 47.00, 37.21, 31.17, 28.94, 19.72, 18.34. HRMS (ESI) calcd for C_33_H_40_N_3_O_5_ 558.2962 [M + H]^+^, found 558.2958.

#### 4.1.3. Synthesis of (9H-fluoren-9-yl)methyl ((S)-1-(((S)-1-(((S)-1-amino-3-methyl-1-oxobutan-2-yl)amino)-3-(4-(tert-butoxy)phenyl)-1-oxopropan-2-yl)amino)-3-(4-(tert-butoxy)phenyl)-1-oxopropan-2-yl)carbamate (**12**)

To a solution of HNEt_2_ (1 mL) and MeCN (4 mL), compound **10** (905 mg, 2.7 mmol) was added and the solution was stirred at room temperature for 1 h. After the reaction was completed, the solution was concentrated to remove the solvent. Then, the residue was dissolved in 20 mL of dry CH_2_Cl_2_ and Fmoc-Tyr(OtBu)-OH (7) (1.2 g, 2.7 mmol) was added. The mixture solution was cooled to 0 °C with an ice bath. HOBt (364 mg, 2.7 mmol), HBTU (2.0 g, 5.4 mmol), and DIPEA (2 mL, 12 mmol) were added to the solution at 0 °C. Then, the ice bath was removed, and the mixture solution was stirred at room temperature overnight. After the reaction was completed (detected by TLC), the mixture was washed with 1 N NaHSO_4_, and saturated in NaHCO_3_ and brine. After drying over anhydrous Na_2_SO_4_, the solution was concentrated and the residue was purified with silica gel column (CH_2_Cl_2_/MeOH = 50/1) to obtain compound **12** (1.6 g, 78%) as a white solid. ^1^H NMR (300 MHz, DMSO-d6) δ 8.19 (d, *J* = 8.1 Hz, 1H), 7.87 (d, *J* = 7.5 Hz, 2H), 7.77 (d, *J* = 8.9 Hz, 1H), 7.64 (d, *J* = 7.4 Hz, 2H), 7.55 (d, *J* = 8.9 Hz, 1H), 7.46–7.25 (m, 5H), 7.19–7.03 (m, 5H), 6.79 (t, *J* = 8.1 Hz, 4H), 4.29–3.95 (m, 5H), 3.62 (td, *J* = 6.6, 4.0 Hz, 1H), 3.19–2.97 (m, 2H), 2.84 (dt, *J* = 12.6, 5.3 Hz, 2H), 2.04–1.91 (m, 1H), 1.18 (d, *J* = 6.5 Hz, 18H), 0.84 (t, *J* = 6.4 Hz, 6H). ^13^C NMR (75 MHz, DMSO) δ 173.13, 171.88, 171.26, 166.61, 156.08, 153.82, 144.16, 141.11, 133.06, 132.71, 130.15, 128.06, 127.48, 125.72, 123.80, 123.65, 120.49, 77.97, 66.19, 57.83, 56.71, 54.28, 54.07, 46.99, 42.32, 37.04, 31.07, 28.93, 19.72, 18.53, 18.27, 17.16, 12.92. HRMS (ESI) calcd for C_46_H_57_N_4_O_7_ 777.4221 [M + H]^+^, found 777.4218.

#### 4.1.4. Synthesis of tert-butyl (*S*)-4-((((9*H*-fluoren-9-yl)methoxy)carbonyl)amino)-5-(((*S*)-1-(((*S*)-1-(((*S*)-1-amino-3-methyl-1-oxobutan-2-yl)amino)-3-(4-(tert-butoxy)phenyl)-1-oxopropan-2-yl)amino)-3-(4-(tert-butoxy)phenyl)-1-oxopropan-2-yl)amino)-5-oxopentanoate (**15**)

To a solution of HNEt_2_ (1 mL) and MeCN (4 mL), compound **12** (714 mg, 1.3 mmol) was added and the solution was stirred at room temperature for 1 h. After the reaction was completed, the solution was concentrated to remove the solvent. Then, the residue was dissolved in 20 mL of dry CH_2_Cl_2_ and Fmoc-Glu(O^t^Bu)-OH (**14**) (552 mg, 1.3 mmol) was added. The mixture solution was cooled to 0 °C with an ice bath. HOBt (199 mg, 1.3 mmol), HBTU (985 mg, 2.6 mmol), and DIPEA (1 mL, 6 mmol) were added to the solution at 0 °C. Then, the ice bath was removed, and the mixture solution was stirred at room temperature overnight. After the reaction was completed (detected by TLC), the mixture was washed with 1 N NaHSO_4_, an saturated in NaHCO_3_ and brine. After drying over anhydrous Na_2_SO_4_, the solution was concentrated and the residue was purified with silica gel column (CH_2_Cl_2_/MeOH = 20/1) to obtain compound **15** (975 mg, 77%) as a white solid. ^1^H NMR (300 MHz, DMSO-*d*_6_) *δ* 8.24 (d, *J* = 7.7 Hz, 1H), 8.07–7.54 (m, 7H), 7.51–7.21 (m, 6H), 7.09 (dd, *J* = 19.4, 8.5 Hz, 5H), 6.78 (dd, *J* = 25.5, 7.9 Hz, 3H), 4.53 (d, *J* = 26.4 Hz, 2H), 4.22 (td, *J* = 30.5, 29.7, 13.2 Hz, 4H), 3.95 (s, 1H), 3.02–2.67 (m, 3H), 2.20–1.63 (m, 4H), 1.37 (s, 9H), 1.28–0.96 (m, 20H), 0.83 (d, *J* = 7.5 Hz, 6H). ^13^C NMR (75 MHz, DMSO) *δ* 174.53, 173.13, 172.63, 171.39, 171.29, 153.83, 132.81, 132.55, 130.26, 130.12, 123.78, 123.63, 79.81, 78.03, 77.96, 57.86, 54.55, 54.40, 53.53, 31.79, 31.05, 30.42, 29.00, 28.20, 19.72, 18.32. HRMS (ESI) calcd for C_55_H_72_N_5_O_10_ 962.5273 [M + H]^+^, found 962.5270.

#### 4.1.5. Synthesis of methyl (5*S*,8*S*,11*S*,14*S*)-5-(3-(tert-butoxy)-3-oxopropyl)-8,11-bis(4-(tert-butoxy)benzyl)-1-(9*H*-fluoren-9-yl)-14-isopropyl-3,6,9,12-tetraoxo-2-oxa-4,7,10,13-tetraazapentadecan-15-oate (**16**)

(*S*)-2-((((9*H*-fluoren-9-yl)methoxy)carbonyl)amino)-3-(4-(*tert*-butoxy)phenyl)propanoic acid (**7**) (1.4 g, 3 mmol) and methyl *L*-valinate (**9**) (504 mg, 3 mmol) were dissolved in 20 mL of CH_2_Cl_2_ and the mixture solution was cooled to 0 °C with an ice bath. HOBt (405 mg, 3 mmol), HBTU (2.3 mg, 6 mmol), and DIPEA (2 mL, 12 mmol) were added to the solution at 0 °C. Then, the ice bath was removed, and the mixture solution was stirred at room temperature overnight. After the reaction was completed (detected by TLC), the mixture was washed with 1 N NaHSO_4_, and saturated in NaHCO_3_ and brine. After drying over anhydrous Na_2_SO_4_, the solution was concentrated and purified with silica gel column (CH_2_Cl_2_/MeOH = 100/1) to obtain **11** (1.6 g, 91%) as a white solid.

To a solution of HNEt_2_ (0.1 mL) and MeCN (0.4 mL), compound **11** (1.7 g, 3 mmol) was added and the solution was stirred at room temperature for 1 h. After the reaction was completed, the solution was concentrated to remove the solvent. Then, the residue was dissolved in 20 mL of dry CH_2_Cl_2_ and Fmoc-Tyr(O^t^Bu)-OH (**7**) (1.3 g, 3 mmol) was added. The mixture solution was cooled to 0 °C with an ice bath. HOBt (459 mg, 3 mmol), HBTU (2.3 g, 6 mmol), and DIPEA (2 mL, 12 mmol) were added to the solution at 0 °C. Then, the ice bath was removed, and the mixture solution was stirred at room temperature overnight. After the reaction was completed (detected by TLC), the mixture was washed with 1 N NaHSO_4_, and saturated in NaHCO_3_ and brine. After drying over anhydrous Na_2_SO_4_, the solution was concentrated and the residue was purified with silica gel column (CH_2_Cl_2_/MeOH = 50/1) to obtain compound **13** (1.7 g, 73%) as a white solid. 

To a solution of HNEt_2_ (0.1 mL) and MeCN (0.4 mL), compound **13** (2.4 g, 3 mmol) was added and the solution was stirred at room temperature for 1 h. After the reaction was completed, the solution was concentrated to remove the solvent. Then, the residue was dissolved in 20 mL of dry CH_2_Cl_2_ and Fmoc-Glu(O^t^Bu)-OH (**14**) (1.3 g, 3 mmol) was added. The mixture solution was cooled to 0 °C with an ice bath. HOBt (459 mg, 3 mmol), HBTU (2.3 g, 6 mmol), and DIPEA (2 mL, 12 mmol) were added to the solution at 0 °C. Then, the ice bath was removed, and the mixture solution was stirred at room temperature overnight. After the reaction was completed (detected by TLC), the mixture was washed with 1 N NaHSO_4_, and saturated in NaHCO_3_ and brine. After drying over anhydrous Na_2_SO_4_, the solution was concentrated and the residue was purified with silica gel column (CH_2_Cl_2_/MeOH = 50/1) to obtain compound **16** (2.2 g, 74%) as a white solid. ^1^H NMR (300 MHz, Methanol-*d*_4_) *δ* 7.82 (d, *J* = 7.5 Hz, 2H), 7.67 (d, *J* = 7.4 Hz, 2H), 7.46–7.24 (m, 4H), 7.23–7.03 (m, 5H), 7.00–6.72 (m, 6H), 4.32 (ddt, *J* = 20.7, 14.7, 7.2 Hz, 4H), 4.05 (dd, *J* = 8.9, 5.4 Hz, 1H), 3.69 (s, 3H), 3.06 (dd, *J* = 13.5, 5.5 Hz, 2H), 2.86 (s, 2H), 2. 28–2.09 (m, 3H), 1.86 (ddt, *J* = 38.6, 14.5, 7.2 Hz, 2H), 1.46 (s, 9H), 1.32–1.30 (m, 9H), 1.25 (s, 9H), 0.94 (dd, *J* = 7.1, 3.2 Hz, 6H). ^13^C NMR (75 MHz, MeOD) *δ* 172.54, 172.49, 171.80, 171.69, 171.51, 166.60, 153.90, 143.93, 143.72, 141.20, 141.18, 131.80, 129.58, 129.51, 129.46, 129.41, 127.41, 126.78, 124.88, 124.85, 124.13, 123.94, 123.79, 123.75, 119.54, 80.43, 78.05, 77.98, 66.76, 57.87, 54.48, 51.06, 36.79, 36.48, 31.21, 30.52, 27.79, 26.99, 26.77, 18.03, 17.30. HRMS (ESI) calcd for C_56_H_73_N_4_O_11_ 977.5270 [M + H]^+^, found 977.5266.

#### 4.1.6. Synthesis of tert-butyl (*S*)-4-acetamido-5-(((*S*)-1-(((*S*)-1-(((*S*)-1-amino-3-methyl-1-oxobutan-2-yl)amino)-3-(4-(tert-butoxy)phenyl)-1-oxopropan-2-yl)amino)-3-(4-(tert-butoxy)phenyl)-1-oxopropan-2-yl)amino)-5-oxopentanoate (**17**)

To a solution of HNEt_2_ (0.1 mL) and MeCN (0.4 mL), compound **15** (50 mg, 0.05 mmol) was added and the solution was stirred at room temperature for 1 h. After the reaction was completed, the solution was concentrated to remove the solvent. Then, the compound was dissolved in 5 mL of CH_2_Cl_2_ and the solution was cooled to 0 °C with an ice bath. Then, Et_3_N (11 mg, 0.1 mmol) and acetyl chloride (7.9 mg, 0.1 mmol) were added. The mixture was stirred at room temperature overnight. The solution was diluted with 20 mL of CH_2_Cl_2_ and washed with 1 N NaHSO_4_, saturated in NaHCO_3_ and brine. After drying over anhydrous Na_2_SO_4,_ the solution was concentrated and the residue was purified with silica gel column (CH_2_Cl_2_/CH_3_OH = 50/1 to 20/1) to obtain compound **17** (26 mg, 66%) as a white solid. ^1^H NMR (300 MHz, Methanol-*d*_4_) *δ* 8.14 (s, 1H), 7.92 (dd, *J* = 21.0, 7.5 Hz, 1H), 7.12 (dd, *J* = 25.2, 8.0 Hz, 4H), 6.90 (dd, *J* = 14.9, 8.0 Hz, 4H), 4.60 (d, *J* = 27.3 Hz, 2H), 4.21 (dd, *J* = 19.2, 11.7 Hz, 2H), 3.48 (s, 1H), 3.19–2.80 (m, 4H), 2.17 (dt, *J* = 48.6, 7.0 Hz, 3H), 1.97 (s, 3H), 1.82 (dd, *J* = 16.5, 9.0 Hz, 1H), 1.38 (d, *J* = 41.4 Hz, 27H), 0.97 (t, *J* = 5.9 Hz, 6H). ^13^C NMR (75 MHz, MeOD) *δ* 174.33, 172.37, 172.26, 172.09, 171.77, 153.96, 131.95, 131.74, 129.47, 129.42, 123.85, 123.70, 80.40, 78.10, 78.01, 58.54, 54.88, 54.64, 52.90, 36.46, 31.15, 30.42, 29.25, 27.82, 26.97, 26.63, 21.12, 18.35, 17.21. HRMS (ESI) calcd for C_42_H_64_N_5_O_9_ 782.4698 [M + H]^+^, found 782.4692.

#### 4.1.7. Synthesis of methyl (2*S*,5*S*,8*S*,11*S*)-11-(3-(tert-butoxy)-3-oxopropyl)-5,8-bis(4-(tert-butoxy)benzyl)-2-isopropyl-4,7,10,13-tetraoxo-3,6,9,12-tetraazatetradecanoate (**18**)

To a solution of HNEt_2_ (0.1 mL) and MeCN (0.4 mL), compound **16** (50 mg, 0.05 mmol) was added and the solution was stirred at room temperature for 1 h. After the reaction was completed, the solution was concentrated to remove the solvent. Then, the compound was dissolved in 5 mL of CH_2_Cl_2_ and the solution was cooled to 0 °C with an ice bath. Then, Et_3_N (11 mg, 0.1 mmol) and acetyl chloride (7.9 mg, 0.1 mmol) were added. The mixture was stirred at room temperature overnight. The solution was diluted with 20 mL of CH_2_Cl_2_ and washed with 1 N NaHSO_4_, saturated in NaHCO_3_ and brine. After drying over anhydrous Na_2_SO_4_, the solution was concentrated and the residue was purified with silica gel column (CH_2_Cl_2_/CH_3_OH = 50/1 to 20/1) to obtain compound **18** (31 mg, 78%) as a white solid. ^1^H NMR (300 MHz, Methanol-*d*_4_) *δ* 7.23–7.04 (m, 4H), 6.96–6.80 (m, 4H), 4.64 (ddd, *J* = 26.2, 8.4, 5.9 Hz, 2H), 4.31 (dt, *J* = 5.8, 4.0 Hz, 2H), 3.70 (s, 3H), 3.14–2.80 (m, 4H), 2.26 (t, *J* = 7.7 Hz, 2H), 2.18–2.07 (m, 1H), 1.96 (s, 3H), 1.89–1.74 (m, 1H), 1.46 (d, *J* = 2.3 Hz, 9H), 1.32 (d, *J* = 2.1 Hz, 18H), 0.95 (dd, *J* = 6.9, 3.7 Hz, 6H). ^13^C NMR (75 MHz, MeOD) *δ* 172.39, 172.06, 171.96, 171.82, 171.70, 171.54, 153.94, 131.81, 131.76, 129.73, 129.51, 129.46, 123.91, 123.77, 123.69, 80.37, 78.02, 77.98, 57.85, 54.48, 54.43, 52.67, 51.07, 36.85, 36.63, 31.18, 30.55, 27.83, 27.78, 26.98, 26.93, 26.76, 21.13, 18.05, 17.32. HRMS (ESI) calcd for C_43_H_65_N_4_O_10_ 797.4695 [M + H]^+^, found 797.4692.

#### 4.1.8. Synthesis of tert-butyl (S)-5-(((S)-1-(((S)-1-(((S)-1-amino-3-methyl-1-oxobutan-2-yl)amino)-3-(4-(tert-butoxy)phenyl)-1-oxopropan-2-yl)amino)-3-(4-(tert-butoxy)phenyl)-1-oxopropan-2-yl)amino)-4-benzamido-5-oxopentanoate (**19**)

Compound **19** (42 mg, 70%) was synthesized by a procedure similar to that used to prepare compound **17** as a white solid. ^1^H NMR (300 MHz, Methanol-*d*_4_) *δ* 7.87 (d, *J* = 7.6 Hz, 2H), 7.65–7.43 (m, 3H), 7.10 (dd, *J* = 23.3, 8.0 Hz, 4H), 6.83 (dd, *J* = 34.4, 8.1 Hz, 4H), 4.70–4.47 (m, 3H), 4.20 (d, *J* = 7.3 Hz, 1H), 3.17–2.80 (m, 4H), 2.34 (t, *J* = 7.2 Hz, 2H), 2.09 (ddt, *J* = 25.1, 18.1, 8.2 Hz, 3H), 1.44 (s, 9H), 1.28 (d, *J* = 16.7 Hz, 18H), 0.97 (dd, *J* = 6.8, 4.0 Hz, 6H). ^13^C NMR (75 MHz, MeOD) *δ* 174.37, 172.72, 172.38, 171.85, 171.79, 168.89, 153.92, 133.45, 131.94, 131.70, 131.63, 129.45, 129.38, 128.19, 127.27, 123.85, 123.73, 80.53, 78.10, 77.94, 58.56, 54.87, 54.77, 53.71, 36.56, 36.48, 31.37, 30.42, 27.80, 26.96, 26.26, 18.36, 17.21. HRMS (ESI) calcd for C_47_H_66_N_5_O_9_ 844.4855 [M + H]^+^, found 844.4855.

#### 4.1.9. Synthesis of tert-Butyl (*S*)-4-(3-((3R,5R,7R)-adamantan-1-yl)ureido)-5-(((*S*)-1-(((*S*)-1-(((*S*)-1-amino-3-methyl-1-oxobutan-2-yl)amino)-3-(4-(tert-butoxy)phenyl)-1-oxopropan-2-yl)amino)-3-(4-(tert-butoxy)phenyl)-1-oxopropan-2-yl)amino)-5-oxopentanoate (**20**)

Compound **20** (35 mg, 56%) was synthesized by a procedure similar to that used to prepare compound **17** as a white solid. ^1^H NMR (300 MHz, Methanol-*d*_4_) *δ* 7.12 (dd, *J* = 25.8, 8.0 Hz, 4H), 6.90 (dd, *J* = 13.5, 8.0 Hz, 4H), 4.66 (d, *J* = 7.8 Hz, 1H), 4.49 (d, *J* = 7.1 Hz, 1H), 4.20 (d, *J* = 7.6 Hz, 1H), 4.04 (t, *J* = 6.9 Hz, 1H), 3.19–2.84 (m, 4H), 2.32–1.82 (m, 13H), 1.70 (s, 7H), 1.46 (s, 9H), 1.32 (s, 18H), 0.97 (dd, *J* = 6.8, 3.7 Hz, 6H). ^13^C NMR (75 MHz, MeOD) *δ* 174.45, 173.73, 172.41, 171.91, 166.61, 157.98, 153.93, 132.11, 131.64, 129.40, 123.85, 80.34, 77.97, 58.74, 55.07, 54.71, 53.35, 50.25, 41.94, 36.61, 36.10, 31.11, 30.37, 29.57, 27.85, 27.79, 27.47, 26.98, 18.37, 17.26. HRMS (ESI) calcd for C_51_H_77_N_6_O_9_ 917.5752 [M + H]^+^, found 917.5746.

#### 4.1.10. Synthesis of tert-Butyl (4*S*)-4-((1*S*,3*R*,5*S*)-adamantane-1-carboxamido)-5-(((*S*)-1-(((*S*)-1-(((*S*)-1-amino-3-methyl-1-oxobutan-2-yl)amino)-3-(4-(tert-butoxy)phenyl)-1-oxopropan-2-yl)amino)-3-(4-(tert-butoxy)phenyl)-1-oxopropan-2-yl)amino)-5-oxopentanoate (**21**)

To a solution of HNEt_2_ (1 mL) and MeCN (4 mL), compound **15** (68 mg, 0.07 mmol) was added and the solution was stirred at room temperature for 1 h. After the reaction was completed, the solution was concentrated to remove the solvent. Then, the residue was dissolved in 2 mL of dry CH_2_Cl_2_ and 1-adamantanecarboxylic acid (12 mg, 0.07 mmol) was added. The mixture solution was cooled to 0 °C with an ice bath. HOBt (9 mg, 0.07 mmol), HBTU (50 mg, 0.14 mmol), and DIPEA (22 mg, 0.14 mmol) were added to the solution at 0 °C. Then, the ice bath was removed, and the mixture solution was stirred at room temperature overnight. After the reaction was completed (detected by TLC), the mixture was washed with 1 N NaHSO_4_, and saturated in NaHCO_3_ and brine. After drying over anhydrous Na_2_SO_4_, the solution was concentrated and the residue was purified with silica gel column (CH_2_Cl_2_/MeOH = 50/1) to obtain compound **21** (38 mg, 61%) as a white solid. ^1^H NMR (300 MHz, Methanol-*d*_4_) *δ* 7.12 (dd, *J* = 25.3, 8.2 Hz, 4H), 6.90 (dd, *J* = 15.6, 8.3 Hz, 4H), 4.66 (dd, *J* = 8.2, 5.9 Hz, 1H), 4.54 (dd, *J* = 8.5, 5.2 Hz, 1H), 4.34–4.13 (m, 2H), 3.74 (p, *J* = 6.6 Hz, 1H), 3.15–2.82 (m, 4H), 2.27 (t, *J* = 7.2 Hz, 1H), 2.03 (s, 3H), 1.86–1.74 (m, 8H), 1.46 (s, 9H), 1.39 (d, *J* = 6.6 Hz, 7H), 1.32 (s, 18H), 0.97 (t, *J* = 6.1 Hz, 6H). ^13^C NMR (75 MHz, MeOD) *δ* 179.52, 174.35, 172.84, 172.42, 171.80, 171.70, 154.08, 153.96, 131.91, 131.50, 129.46, 129.44, 123.86, 123.74, 80.54, 78.09, 77.94, 58.57, 54.79, 54.55, 54.45, 52.97, 42.39, 40.47, 38.62, 36.54, 36.10, 31.26, 30.43, 28.18, 27.85, 27.82, 27.00, 26.28, 18.36, 17.24, 15.87, 11.74. HRMS (ESI) calcd for C_51_H_76_N_5_O_9_ 902.5637 [M + H]^+^, found 902.5632.

#### 4.1.11. Synthesis of tert-Butyl (*S*)-5-(((*S*)-1-(((*S*)-1-(((*S*)-1-amino-3-methyl-1-oxobutan-2-yl)amino)-3-(4-(tert-butoxy)phenyl)-1-oxopropan-2-yl)amino)-3-(4-(tert-butoxy)phenyl)-1-oxopropan-2-yl)amino)-4-decanamido-5-oxopentanoate (**22**)

Compound **22** (36 mg, 58%) was synthesized by a procedure similar to that used to prepare compound **21** as a white solid. ^1^H NMR (300 MHz, Methanol-*d*_4_) *δ* 7.11 (dd, *J* = 27.5, 8.0 Hz, 4H), 6.89 (dd, *J* = 16.9, 8.0 Hz, 4H), 4.68–4.50 (m, 2H), 4.24 (dd, *J* = 28.9, 7.0 Hz, 2H), 3.20–2.76 (m, 5H), 2.36–1.78 (m, 7H), 1.38 (d, *J* = 41.7 Hz, 40H), 0.94 (dt, *J* = 19.6, 6.4 Hz, 9H). ^13^C NMR (75 MHz, MeOD) *δ* 175.10, 174.35, 172.36, 171.79, 171.74, 153.99, 153.95, 131.96, 131.67, 129.47, 129.41, 123.86, 123.73, 80.40, 78.10, 77.97, 58.55, 54.89, 54.67, 52.80, 36.59, 36.44, 35.42, 31.64, 31.15, 30.42, 29.22, 29.08, 29.02, 28.97, 27.85, 27.82, 26.98, 26.56, 25.46, 22.31, 18.36, 17.23, 13.03. HRMS (ESI) calcd for C_50_H_80_N_5_O_9_ 894.5950 [M + H]^+^, found 894.5956.

#### 4.1.12. Synthesis of tert-Butyl (*S*)-5-(((*S*)-1-(((*S*)-1-(((*S*)-1-amino-3-methyl-1-oxobutan-2-yl)amino)-3-(4-(tert-butoxy)phenyl)-1-oxopropan-2-yl)amino)-3-(4-(tert-butoxy)phenyl)-1-oxopropan-2-yl)amino)-4-(cyclohexanecarboxamido)-5-oxopentanoate (**23**)

Compound **23** (38 mg, 64%) was synthesized by a procedure similar to that used to prepare compound **21** as a white solid. ^1^H NMR (300 MHz, Methanol-*d*_4_) *δ* 7.12 (dd, *J* = 28.0, 8.1 Hz, 4H), 6.89 (dd, *J* = 17.8, 8.1 Hz, 4H), 4.70–4.50 (m, 2H), 4.31–4.15 (m, 2H), 3.17–2.81 (m, 4H), 2.39–1.61 (m, 11H), 1.45 (s, 10H), 1.32 (s, 22H), 0.97 (t, *J* = 6.1 Hz, 6H). ^13^C NMR (75 MHz, MeOD) *δ* 177.96, 174.35, 172.44, 172.37, 171.80, 171.70, 154.03, 153.95, 131.93, 131.57, 129.44, 123.85, 123.70, 80.43, 78.10, 77.95, 58.56, 54.84, 54.58, 52.69, 44.62, 36.58, 36.52, 31.19, 30.42, 29.44, 29.13, 27.85, 27.82, 26.98, 26.60, 25.46, 25.38, 25.32, 18.36, 17.23. HRMS (ESI) calcd for C_47_H_72_N_5_O_9_ 850.5330 [M + H]^+^, found 850.5326.

#### 4.1.13. Synthesis of tert-Butyl (*S*)-5-(((*S*)-1-(((*S*)-1-(((*S*)-1-amino-3-methyl-1-oxobutan-2-yl)amino)-3-(4-(tert-butoxy)phenyl)-1-oxopropan-2-yl)amino)-3-(4-(tert-butoxy)phenyl)-1-oxopropan-2-yl)amino)-4-(((benzyloxy)carbonyl)amino)-5-oxopentanoate (**24**)

Compound **24** (36 mg, 61%) was synthesized by a procedure similar to that used to prepare compound **17** as a white solid. ^1^H NMR (300 MHz, Methanol-d_4_) δ 7.49–7.23 (m, 5H), 7.17–7.06 (m, 4H), 6.94–6.84 (m, 4H), 5.21–5.02 (m, 2H), 4.59 (ddd, *J* = 13.9, 8.5, 3.5 Hz, 2H), 4.08 (dd, *J* = 8.9, 5.5 Hz, 1H), 3.22–2.78 (m, 5H), 2.25 (t, *J* = 7.5 Hz, 2H), 2.16–2.05 (m, 1H), 1.98–1.75 (m, 2H), 1.44 (s, 9H), 1.31 (d, *J* = 3.2 Hz, 18H), 1.00–0.94 (m, 6H). ^13^C NMR (75 MHz, MeOD) δ 174.33, 172.66, 172.46, 171.74, 153.94, 136.58, 131.92, 131.74, 129.46, 129.43, 128.09, 127.64, 127.45, 123.87, 123.76, 80.42, 78.13, 78.02, 66.50, 58.53, 54.85, 54.60, 36.52, 31.16, 30.43, 27.81, 26.96, 18.35, 17.20. HRMS (ESI) calcd for C_48_H_68_N_5_O_10_ 874.4960 [M + H]^+^, found 874.4956.

### 4.2. Cell Culture

HEK293 cells were maintained in Dulbecco’s Modified Eagle Medium (DMEM) and F-12 (Invitrogen, Carlsbad, CA, USA), supplemented with 0.05% glucose, 0.5 mM pyruvate, 10% fetal bovine serum, 100 units/mL penicillin, and 100 µg/mL streptomycin (Invitrogen), and incubated at 37 °C with 5% CO_2_. The double stable HEK293 cell line expressing CD4-Nav1.6C-tail-NLuc and CLuc-FGF14 used for LCA experiments was described in a previous study [33] and was maintained using selective antibiotics (0.5 mg/mL G418 and 5 µg/mL puromycin). HEK293 cells stably expressing the human Nav1.6 channel (hereafter referred to as HEK-Nav1.6 cells) were maintained similarly, except for the addition of 500 μg/mL G418 (Invitrogen) to maintain stable Nav1.6 expression. Cells were transfected at 80–90% confluence with equal amount (1 μg each) of plasmid pairs using Lipofectamine 3000 (Invitrogen, Waltham, MA, USA) according to the manufacturer’s instructions. HEK-Nav1.6 cells were washed and re-plated at very low density prior to electrophysiological recordings [31,33,36,45].

### 4.3. Split-Luciferase Complementation Assay (LCA)

Cells were trypsinized (0.25%), triturated in a medium, and seeded in white, clear-bottom CELLSTAR μClear^®^ 96-well tissue culture plates (Greiner Bio-One, Monroe, NC, USA) at ~0.9 × 10^5^ cells per well in 50 μL of serum-free, phenol red-free DMEM/F12 medium (Invitrogen, Waltham, MA, USA). For transiently transfected cells, the trypsinization occurred 48 h post-transfection. The cells were incubated for 2 h, followed by addition of 50 µL of serum-free medium containing either 0.5% DMSO alone (vehicle; *n* = 32 replicates per plate) or compounds (25 µM for screening and 0.1–175 µM for dose responses; *n* = 4 replicates per treatment condition per plate) dissolved to a final concentration of 0.5% DMSO, with a minimum of 3 independent experiments per compound. Following 12 h incubation at 37 °C, the reporter reaction was initiated by injection of 100 μL substrate solution containing 1.5 mg/mL of d-luciferin (final concentration = 0.75 mg/mL) by the Synergy™ H1 Multi-Mode Microplate Reader (BioTek, Winooski, VT, USA). Luminescence readings (integration time 0.5 s) were performed at 2-min intervals for 20 min, and the cells were maintained at 37 °C throughout the measurements. Signal intensity for each well was calculated as a mean value of peak luminescence. The calculated values were expressed as percentage of mean signal intensity of the per plate control samples. Detailed methods for LCA can be found in previous studies [31,32,33,36,46,47].

Dose response curves were obtained using GraphPad Prism 8 by fitting the data with a non-linear regression:(1)$$A+\frac{B-A}{1+10^{log\left(x_{0}-x\right)H}}$$
where *x* is log_10_ of the compound concentration in M, *x_0_* is the inflection point (EC_50_ or IC_50_), *A* is the bottom plateau effect, *B* is the top plateau effect, and *H* is the Hill slope. Potency (IC_50_/EC_50_) and efficacy (percent luminescence at the bottom or top plateau for inhibitors (*I_Min_*) or enhancers (*E_Max_*), respectively) were calculated from the dose response non-linear regression.

### 4.4. Cell Viability Assay

The CellTiter-Blue^®^ Cell Viability (CTB) assay (Promega, Madison, WI, USA) was used to counter-screen top compounds. Immediately following LCA luminescence reading from cells treated with experimental compounds or Tamoxifen (positive control with known toxicity toward HEK293 cells [33]), 30 µL of 1X CTB reagent was dispensed into 96-well plates, plates were incubated overnight (16 h) at 37 °C, and fluorescence was detected using the Synergy H1 reader (excitation λ = 560 nm, emission λ = 590 nm). Cell viability was expressed as percent of the mean fluorescent signal intensity of on-plate negative controls.

### 4.5. Protein Expression and Purification

The two plasmids (pET28a-FGF14; pET30a-Nav1.6) for protein expression and purification of FGF14 (accession number NP_787125; aa 64–252) and Nav1.6 C-tail (accession number #NP_001171455; aa 1763–1912) have been previously described [36,45] and were transformed into *E. coli* BL21 (DE3) pLys (Invitrogen). Cells were grown until OD_600_ = 0.7, and the recombinant proteins were expressed after induction with 0.1 mM isopropyl thio-*β*-d-galacto-pyranoside (IPTG) for 24 h at 16 °C. Cells were harvested and lysed by sonication at 4 °C in lysis/binding buffer containing the following components (mM): 10 sodium phosphate (prepared from 0.5 M of Na_2_HPO_4_ and NaH_2_PO_4_), 25 4-(2-hydroxyethyl)-1-piperazineethanesulfonic acid (HEPES), 150 NaCl, phenyl methyl sulphonyl fluoride (PMSF) 0.1, 3-((3-cholamidopropyl) dimethylammonio)-1-propanesulfonate (CHAPS) 0.1% pH 7.0 (for FGF14), and with glycerol 10% (for Nav1.6 C-tail) pH 7.5. The respective proteins were centrifuged at 40,000× *g* for 1 h at 4 °C. For purification of FGF14, the supernatant was applied to pre-equilibrated heparin and the proteins were then eluted with NaCl 0.2–2.0 M (sodium phosphate 10 mM, NaCl 0.2–2.0 M, pH 7.0) buffer. For purification of Nav1.6 C-tail, the supernatant was first applied to a cobalt column (Thermo Fisher Scientific) and eluted with imidazole (200 mM). The Na_v_1.6 C-tail was further purified using a HiTrap Q Sepharose Fast Flow column (GE Healthcare, Pittsburgh, PA, USA) using a buffer containing Tris-HCl 50 mM and eluted with NaCl (10–500 mM) at pH 7.5. Finally, all concentrated proteins were purified on an AKTA^TM^ fast protein liquid chromatography (FPLC) using a Superdex 200 HiLoad 16 × 60 column and equilibrated in Tris-HCl 50 mM + NaCl 150 mM, pH 7.5 (GE Healthcare Bio-Sciences, Chicago, IL, USA). Protein concentrations were determined using UV absorbance with a Thermo NANODROP.

### 4.6. Surface Plasmon Resonance Spectroscopy

SPR experiments were performed on a Biacore T100 instrument (GE Healthcare, Chicago, IL, USA). Proteins were immobilized on CM5 sensor chips using 10 mM sodium acetate buffer (pH 5.5) with the Amine Coupling Kit (GE Healthcare), as per the manufacturer’s instructions, to a final RU value of 17,896 for FGF14 and 6,800 for Nav1.6 C-tail. No protein was coupled to the control flow channels of the chip (Lanes 1 and 3). The interaction of experimental compounds against FGF14 and Nav1.6 proteins were studied at 25 °C using a flow rate of 60 µL/min. Compounds were serially diluted (0.195–200 µM) in PBS supplemented with Tween-20 0.005% and 2% DMSO. Each sample was injected over the chip for 120 s followed by a dissociation period of 150 s, and finally, chip surface regeneration (600 mM NaCl, 5% DMSO) for 120 s. Each compound was tested over at least two independent experiments with concentrations of 0.195, 0.39, 0.78, 1.56, 3.12, 6.25, 12.5, 25, 50, 100, and 200 µM, including 12.5 and 25 µM, and blanks (buffer alone) in duplicate. A DMSO calibration was performed for each experiment (1.5–2.8% (*v*/*v*) DMSO) to correct for bulk refractive index shifts [48]. For each compound injection, nonspecific responses (“0 μM” solution prepared similarly to experimental compound samples) for each corresponding experiment were subtracted from compound sensorgrams/traces prior to data analysis. Kinetic data were analyzed using the Biacore T100 Analysis software. Following visual inspection of the binding curves, the equilibrium constant (K_D_) was calculated using two methods: (1) maximal responses were plotted against compound concentration, and the steady-state K_D_ was calculated from the fitted saturation binding curve, and (2) a kinetic analysis of each ligand/analyte interaction was obtained by fitting the response data to the simplest Langmuir 1:1 interaction model (K_D_ = k_off_/k_on_). The kinetic constants generated from the fitted binding curves were assessed for accuracy based on the distribution of the residuals (even and near zero to baseline). Compounds failing to achieve saturation of binding over the concentration range tested are reported as >100 μM. Graphs were plotted in GraphPad Prism 8 Software (La Jolla, CA, USA).

### 4.7. Electrophysiology

HEK293 cells stably expressing Nav1.6 (HEK-Nav1.6) were grown in culture as described above. Prior to experimentation, HEK-Nav1.6 cells were dissociated using Gibco TrypLE (Thermo Fisher Scientific, Walthman, MA, USA) and plated at low density onto glass coverslips situated at the base of wells in CELLSTAR^®^ 24-Well Multi-Well Plates (Greiner Bio-One, Monroe, NC, USA). Cells were then incubated on glass cover slips for 2–3 h. After incubation, glass cover slips were transferred to the recording chamber containing 3 mL of extracellular solution comprised of the following salts (mM): 140 NaCl, 3 KCl, 1 MgCl_2_, 1 CaCl_2_, 10 HEPES, 10 glucose, pH 7.3. After incubating for thirty minutes in the extracellular solution containing either DMSO or experimental compound, recordings were then performed at room temperature (20–22 °C) using a MultiClamp 700B amplifier (Molecular Devices, Sunnyvale, CA) and borosilicate glass pipettes (resistance of 3–6 MΩ) containing intracellular solution comprised of the following salts (mM): 130 CH_3_O_3_SCs, 1 ethylene glycol tetraacetic acid (EGTA), 10 NaCl, 10 HEPES, pH 7.3. Dial settings on the amplifier were used to estimate membrane capacitance and series resistance and compensated for electronically by 70–80%. Prior to digitization and storage, data were acquired at 20 kHz and filtered at 5 kHz. Clampex 9.2 software (Molecular Devices), interfaced to the electrophysiological equipment using a Digidata 1200 analog-digital interface (Molecular Devices), was used to control all experimental parameters. To assess the relationship of I_Na_ and voltage, HEK-Nav1.6 cells were evoked by depolarizations to test potentials ranging from –100 to +60 mV from a holding potential of −70 mV, followed by a voltage pre-step pulse of −120 mV. To assess steady-state inactivation of Nav channels, HEK-Nav1.6 cells were evoked using a paired-pulse protocol in which cells were stepped, from the holding potential, to varying test potentials ranging from −120 to +20 mV (pre-pulse) prior to a test pulse to −20 mV. To assess long-term inactivation, HEK-Nav1.6 cells were subjected to four depolarizations at 0 mV for 16 ms separated by three recovery intervals at –90 mV for 40 ms.

### 4.8. Electrophysiology Data Analysis

*I_Na_* was normalized to membrane capacitance to determine current density by dividing I_Na_ amplitude by membrane capacitance. Current density was then plotted as a function of the holding potential to characterize current-voltage relationships. Tau of inactivation was calculated by fitting the decay phase of currents at −10 mV with a one-term exponential function. To assess voltage-dependence of Nav1.6 activation, conductance (*G_Na_*) was first calculated using the following equation:(2)$$G_{Na}=\frac{I_{Na}}{\left(V_{m\;}-E_{rev\;}\right)}$$
where *I_Na_* is current amplitude at voltage *V_m_*, and *E_rev_* is the Na^+^ reversal potential. Steady-state activation curves were then generated by plotting normalized *G_Na_* as a function of test potential. Plotted data was then fitted with the Boltzmann equation to determine V_1/2_ of Nav1.6 activation values using the following equation:(3)$$\frac{G_{Na}}{G_{Na,max}}=1+{\;e}^{V_{a}-E_{m}/k}$$
where *G_Na,Max_* is the maximum conductance, *V_a_* is the membrane potential of half-maximal activation, E_m_ is the membrane voltage, and *k* is the slope factor. To assess steady-state inactivation, *I_Na_* normalized to max *I_Na_ (I_Na_/I_Na,Max_)* at the test potential was plotted as a function of pre-pulse potential. Data was then fitted with the Boltzmann function to determine V_1/2_ using the following equation:(4)$$\frac{I_{Na}}{I_{Na,max}}=\frac{1}{1+e^{V_{h}-E_{m}/k}}$$
where *V_h_* is the potential of half-maximal inactivation, E_m_ is the membrane voltage, and k is the slope factor. For long-term inactivation, the peak I_Na_ of depolarization cycles 2–4 was normalized to the peak I_Na_ of depolarization cycle 1 (*I_Na_/I_Na,Cycle 1_*) and plotted as a function of depolarization cycle. Data analysis was performed using Clampfit 11.1 (Molecular Devices) and GraphPad Prism 8 (La Jolla, CA, USA) software. Results were expressed as mean ± standard error of the mean (SEM). Statistical significance was determined via unpaired t-tests comparing 0.1% DMSO to an experimental group, with *p* < 0.05 being considered statistically significant.

### 4.9. Molecular Docking Method

The molecular docking study was performed using Schrödinger Small-Molecule Drug Discovery Suite (Schrödinger, LLC, New York, NY, USA). FGF14:Nav1.6 homology model [36] was generated using the FGF13:Nav1.5:CaM ternary complex crystal structure (PDB code: 4DCK) as a template. The FGF14:Nav1.6 homology model was prepared with Schrödinger Protein Preparation Wizard using default settings. During this step, hydrogens were added, crystal waters were removed, and partial charges were assigned using the OPLS-2005 force field. The SiteMap (Schrödinger, LLC, New York, NY, USA) calculation was performed and a potential binding site was identified on the PPI of FGF14:Nav1.6. The chain of FGF14 was excluded from the model and the grid center was chosen on Nav1.6 *C*-terminal tail at the previously identified binding site generated from SiteMap results. The grid box was sized in 24 Å on each side to cover the PPI surface on Nav1.6. The 3D structure of ligands **12**, **17**, and **19** were created using Schrödinger Maestro and a low-energy conformation was calculated using LigPrep. Docking was employed with Glide using the SP protocol. Docked poses were incorporated into Schrödinger Maestro for a ligand–receptor interactions visualization. Top ranking of Glide GScore, as well as biological and chemical rationales were used for ligand docked pose evaluation and selection. The top docked poses of ligands **12**, **17**, and **19** were superimposed with FGF14:Nav16 complex homology model for an overlay analysis.

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
