# Peer review of "Bidirectional Modulation of the Voltage-Gated Sodium (Nav1.6) Channel by Rationally Designed Peptidomimetics"

_molecules, 2020, doi:10.3390/molecules25153365_

Round 1

Reviewer 1 Report

The manuscript by Dvorak et al. describes the design, synthesis and subsequent in vitro testing of novel peptidomimetics that modulate Nav1.6 channel. The topic is interesting and important and the results are presented in a reliable manner. I cannot fully judge on the patch-clamp data, because I am not an expert in that field, but this part looks plausible to me.
Given the new data presented in this manuscript and the methods used, I would like to recommend this study for publication in Molecules. However, some points should be addressed prior publication:
1) Could the authors comment on the potential of their peptidomimetics to cross the blood-brain barrier. This would be important, if cmp12 or cmp19 should be applied in in vivo models, or is there an idea how to further modify these compounds?
2) Figure 2B shows 100% cell viability for all tested compounds. It is a very rare occasion that none of the tested compounds showed any effect. What about testing a ‘toxic’ compound as control?
3) The authors state that 12 and 19 are allosteric modulators (NAM, PAM). What is the mechanistic idea of the bidirectional behavior of 12 and 19? Is there an explanation with a link to the structural differences?
4) While I appreciate the usage of molecular modeling, it is hard to directly compare the docking poses in figure 6 (A and C) due to the (slightly) different perspective and different residues showing up (even there is a good reason for the different residues). Maybe a superimposition would be good way to directly compare the two poses.

Reviewer 2 Report

Comprehension of the function and possible regulation of voltage-gated Na+ channels could help to treat various channelopathies. Thus the study of possible disruption of protein-protein interactions which regulated its function is very important. The author prepared and tested various peptidomimetic based analogs with diverse modifications which allowed to identify positive and negative allosteric modulation of such interaction.

The manuscript is written with the logical structure, all experiments were described clearly as well as their conclusions.

With respect to this I suggest to publish these results in present form.

Reviewer 3 Report

This manuscript reports the investigation of peptide derivatives that modulate Nav1.6 function by interfering with intracellular Nav1.6-FGF14 interaction. The 11 new compounds were initially tested on a previously developed split luciferase complementation assay, indicating a decrease of the luminescence signal with one, and an increase with 5 compounds. Concentration-response curves carried out with the same assay on 4 selected compounds showed that the IC50 or EC50 values were in the range of ~10-50 micromolar, and quantified also the maximal changes in luminescence by these compounds. The binding kinetics of these 4 compounds to recombinant FGF14 and to the Nav1.6 terminus were determined, showing binding of these compounds to both proteins with low micromolar affinities. Patch-clamp experiments carried out with two compounds confirmed that one of these compounds potentiated, while the other inhibited Nav1.6 currents. Finally, in silico docking suggests that the compound that inhibits Nav1.6 binds at the same place as FGF14 does on Nav1.6, while the potentiating compounds binds to a distinct site.

Major points

  1. A statistical analysis is carried out for Fig. 2, and then again for the patch-clamp study (Fig. 5). Such an analysis needs also to be carried out for data of tables 2-4, and statements in the text about differences of efficacy, IC50/EC50 or Kd of the compounds need to be based on the results of these tests.

  1. The docking experiments need to be better documented. Information on the model needs to be provided, the sequence homology between the Nav1.6 and Nav1.5 C-terminus that is covered by the model needs to be indicated in the text, the volume to which docking was restricted needs to be indicated in a clear way, and it needs to be indicated how the best poses were selected, and how close in rating they were to alternative poses of the same compounds. Since I had no access to the supplementary figure, it was difficult to estimate how the poses of compounds 12 and 19 relate to each other and to the interaction site of FGF14 on Nav1.6. It is important to indicate how these compounds bind, related to the FGF14-Nav1.6 interaction site. I suggest to include Fig. S1 as regular figure in the manuscript. Based on these structural analyses, can the authors make suggestions on the mechanism by which compound 19 strengthens the FGF14-Nav1.6 interaction and potentiates the Nav1.6 currents?

Specific comments

  1. It seems that these peptidomimetics were initially based on FGF14 structure sequences that interact with the Nav1.6 C-terminus, and that it was expected that they compete with FGF14 for interaction sites on Nav1.6, and interrupt thereby the interaction. From this it is normal to expect that they bind to Nav1.6. However, it seems quite surprising that they also bind to FGF14. Is this because the FGF14 dimerization site overlap with the Nav1.6 interaction site, or what could be the explanation for this? Is it possible that these compounds bind to sites on FGF14 that are very different from the Nav1.6-FGF14 interaction sites, or have the authors arguments for a binding to the Nav1.6-FGF14 interaction site on FGF14? These aspects should be discussed in the text.
  2. For the terms "cLogP" and "tPSA" of table 1 it would be good to include a definition, besides the link.
  3. For the electrophysiology experiments, some additional information on the protocols should be added. Indicate the frequency at which the channels were stimulated ("sweep interval"); indicate also whether a leak cancellation (as P/4 or similar) was used. Indicate also the duration of the -120mV prepulse in activation curves. The protocol of activation curves in Fig. 5A indicates that test pulses of the range of -60 to +60mV were used, however, Fig. 5B and E show data points between -100 and +60mV, consistent also with the description of the protocol in the methods. Thus, the scheme in Fig. 5A needs to be corrected. For the steady-state inactivation (Fig. 5G), the duration of the conditioning pulses needs to be indicated.
  4. In the figure legends, all panels are indicated in lower case letters, while in the figures themselves capital letters are used.
  5. For surface plasmon resonance spectroscopy, the solution flow rate is indicated 50 ul/min in the methods (line 731) and as 60 ul/min in the text (line 210) and the legend of Fig. 4 (line 256).
  6. On line 691, the "H" (Hill coefficient) should be in the exponent.
  7. In a previous study these authors have shown that the related compound ZL 177, which inhibits Nav1.6 currents, accelerates the kinetics of current decay, and induces a depolarizing shift in the voltage dependence of activation. The authors should explain in the text possible reasons for the absence of such changes with compound 12, which induced an even stronger current inhibition than ZL177.
